# Strengthening the Culture of Well-Being in Rural Hospitals Through RISE Peer Support

**DOI:** 10.3390/healthcare14010091

**Published:** 2025-12-30

**Authors:** Mansoor Malik, Gayane Yenokyan, Henry Michtalik, Jane Miller, Cheryl Connors, Christine M. Weston, Kristina Weeks, William Hu, Matt Norvell, Albert W. Wu

**Affiliations:** 1Department of Psychiatry, Johns Hopkins University School of Medicine, Baltimore, MD 21287, USA; 2Department of Biostatistics, Johns Hopkins Bloomberg School of Public Health, Baltimore, MD 21205, USA; 3Department of Medicine, Johns Hopkins School of Medicine, Baltimore, MD 21287, USAawu@jhu.edu (A.W.W.); 4The Armstrong Institute for Patient Safety and Quality, Johns Hopkins School of Medicine, Baltimore, MD 21202, USAmattnorvell@jhmi.edu (M.N.); 5Department of Health Policy and Management, Johns Hopkins Bloomberg School of Public Health, Baltimore, MD 21205, USA; ccicio1@jhmi.edu (C.C.);; 6Department of Anesthesiology and Critical Care Medicine, Johns Hopkins School of Medicine, Baltimore, MD 21287, USA

**Keywords:** burnout, peer support, resilience, rural health

## Abstract

**Highlights:**

**What are the main findings?**
Implementation of the RISE peer support program in two rural hospital systems led to modest reductions in anxiety and burnout and early, significant increases in workforce resilience.Staff reported major improvements in awareness, accessibility, and perceived usefulness of peer support, reflecting a strengthened culture of well-being within rural healthcare settings.

**What are the implications of the main findings?**
Peer support programs such as RISE are feasible, acceptable, and impactful even in resource-limited rural hospitals, offering a scalable model for workforce well-being.Rural health systems can enhance clinician retention, psychological safety, and organizational support culture by adopting structured emotional-support interventions tailored to local context.

**Abstract:**

**Background:** Burnout among healthcare workers (HCWs) threatens workforce stability and patient care, particularly in rural hospitals where staff shortages, limited resources, and professional isolation amplify stress. Peer support interventions have demonstrated promise in urban centers, but their feasibility and impact in rural settings remain underexplored. **Methods:** We implemented and evaluated the Johns Hopkins RISE (Resilience in Stressful Events) peer support program across two rural hospital systems in the Mid-Atlantic United States. Using pre- and post-implementation surveys, we assessed anxiety (GAD-7), burnout (Maslach Burnout Inventory), resilience (CD-RISC), and perceptions of organizational culture of well-being. Linear and logistic regression models adjusted for age, site, and employment duration were used to evaluate outcomes over time. **Results:** A total of 868 respondents participated across three time points. Burnout and anxiety declined modestly post-implementation, while resilience improved initially but was not sustained at 2-year follow-up. Older employees demonstrated lower anxiety and burnout, while mid-career employees (3–10 years of employment) reported significantly higher distress. Importantly, access to peer support and perceived availability of supportive resources improved significantly over time, reflecting growing program integration. **Conclusions:** RISE was adapted successfully in rural hospital settings, with evidence of reduced burnout, lower anxiety, and increased perceived access to peer support. While resilience gains were not sustained, results suggest that a peer support program tailored to each organization can mitigate workforce distress in rural health systems. Addressing implementation and contextual barriers and sustaining organizational commitment are important for long-term impact. Expanding peer support to rural hospitals may improve workforce retention and care delivery in underserved communities.

## 1. Introduction

Burnout among healthcare workers (HCWs) is a major and worsening global concern, characterized by emotional exhaustion, depersonalization, and reduced personal accomplishment [1]. The 2019 National Academy of Medicine report identified burnout as having reached “crisis levels” in the United States, affecting up to 54% of physicians and nurses and over 60% of trainees [2]. These trends intensified during the COVID-19 pandemic and have remained elevated [3]. A post-COVID meta-analysis of 4419 HCWs found that 82% reported moderate to severe emotional exhaustion [4].

Resilience, the capacity to adapt to adversity, varies among HCWs. Although U.S. physicians report higher average resilience than the general workforce, burnout remains prevalent even among highly resilient clinicians [5]. Emotional exhaustion and cynicism dimensions of burnout show correlation with resilience, but correlation with professional self-efficacy is less consistent [6]. Poor mental health is prevalent in healthcare workforce, including depression, anxiety, and post-traumatic stress disorder (PTSD), with 20–40% of HCWs reporting symptoms, particularly in high-stress specialties such as emergency and critical care [7,8]. These findings suggest that individual resilience alone is insufficient to counter structurally driven burnout, highlighting the importance of organizational and system-level interventions.

Rural HCWs face distinct and compounded risks. Although burnout has been less frequently studied in rural settings, existing evidence shows that rural providers operate in resource-limited environments marked by workforce shortages, high patient volumes, limited specialty support, professional isolation, and the need to fulfill multiple roles [9,10,11,12]. These pressures are associated with higher emotional exhaustion and threaten the sustainability of rural healthcare delivery [13,14]. Burnout also directly increases the likelihood of job turnover two- to threefold [15]. In rural hospitals, where staffing pipelines are already fragile, turnover creates a vicious cycle of reduced access, poor quality of care, and negative population health outcomes [16]. Despite these challenges, research on burnout in rural healthcare workers and supportive strategies are limited compared to urban settings, exposing a critical gap in under-standing and addressing this issue [17].

Although, rural clinicians often demonstrate adaptive resilience grounded in community connectedness, clinical autonomy, and a strong sense of purpose in serving underserved populations [18,19], burnout remains a significant concern. Support interventions appropriate for rural hospital settings are essential to mitigate burnout and improve staff well-being [20].

The RISE (Resilience in Stressful Events) peer support program was originally developed at a large academic medical center as a confidential, rapid-response, peer-to-peer intervention following stressful clinical events [21]. The program has been implemented in the US and globally in over 140 healthcare organizations [22]. Prior RISE studies have shown improvements in perceived organizational and peer support, psychological safety, and reductions in distress following adverse events as well as cost savings [23,24].

Critically, no prior studies have evaluated RISE implementation in rural hospitals and there is limited empirical guidance regarding whether peer support models developed in academic centers are feasible, acceptable, or effective in small, resource-constrained rural health systems. This study addresses this gap by examining the implementation of the RISE peer support program in two rural hospital systems in the Mid-Atlantic United States using pre- and post-implementation surveys conducted over two years. We assess changes in burnout, anxiety, resilience, and perceptions of organizational support, while also documenting implementation challenges and contextual adaptations. By directly evaluating peer support in rural hospitals, this work aims to inform scalable, resource-conscious strategies to strengthen workforce well-being, retention, and patient care in underserved communities.

## 2. Methods

The current study is part of a larger project to implement the RISE peer support program in ambulatory clinics, rural hospitals, Federally Qualified Healthcare Centers (FQHCs) and community-based social service organizations (CBOs). RISE is an evidence-based initiative designed to help healthcare professionals cope with stress, emotional trauma, and burnout by fostering resilience and emotional well-being, providing peer support 24 h a day, 7 days a week.

### 2.1. Participants and Procedures

The study design was a cross-sectional, web-based survey of a repeated random sample of HCW from two regional, not-for-profit rural health systems in the Mid Atlantic between August 2022 and December 2024, before and after the implementation of the RISE peer support program.

### 2.2. Survey Sites

The study was conducted at two rural hospital systems in the Mid-Atlantic region, each representing distinct organizational structures and workforce capacities. University of Maryland Shore Regional Hospital (SRH), located in Eastern Maryland, is a rural health network comprising five geographically dispersed care locations and employing approximately 1800 staff. The system provides a broad range of inpatient and outpatient services to a predominantly rural population and faces challenges typical of small regional hospitals, including workforce shortages, resource constraints, and limited access to specialized services. Bayhealth (BH), based in central and southern Delaware, is a rural healthcare system employing approximately 4000 staff across multiple facilities. BH operates two hospitals and several affiliated outpatient sites, offering a wider range of specialty care and infrastructure compared to SRH, while still serving largely rural communities. The differences in scale, staffing, and resources between SRH and BH provided an opportunity to evaluate the feasibility and adaptability of the RISE peer support model in different rural healthcare environments.

### 2.3. Intervention

Previously published RISE protocols were adopted to the unique needs of healthcare workers in rural institutions. Phase 1 focuses on leadership training and ensuring support from key stakeholders, while Phase 2 involves recruitment and training of peer support volunteers, hospital-wide implementation, referral pathways, and promotion of confidential access.

### 2.4. Measures

The measures included in both the pre and post surveys included: the General Anxiety Disorder scale (GAD7) (7 questions) [25], Connor-Davidson Resilience (CD-RISC2) scale (2 questions) [26], the Maslach Burnout Inventory (MBI-2) (2 questions) [27] and other measures of job satisfaction, trust in management, the organization’s commitment to employee health and well-being, and the perceived support from coworkers and supervisors. These included 5 items from the National Institute for Occupational Safety and Health (NIOSH) Worker Well-Being Questionnaire assessing job satisfaction, perceptions of employees’ ability to count on their supervisor and coworkers, trust in the management of the organization, and perceptions of their organization’s commitment to employee health and well-being [28]. We included two questions from the Safety, Communication, Operational Reliability, and Engagement (SCORE) survey related to institutional safety culture and workforce well-being [29].

The research team created new questions when no suitable validated scales existed. New survey items were developed by a multidisciplinary team with expertise in healthcare workforce well-being, peer support, patient safety, and implementation science. These items were iteratively reviewed for content validity, clarity, and relevance. The newly developed items included measures of perceived organizational well-being and resilience (2 items), availability of confidential peer support resources (3 items), and availability of workplace support following stressful events (5 items). Psychometric testing using exploratory and confirmatory factor analysis demonstrated a robust three-factor structure, Organizational Support, Access to Peer Support, and Availability of Support, accounting for 84.9% of total variance, with excellent model fit (CFA RMSEA = 0.049; CFI = 0.992). Internal consistency was high across all domains (Cronbach’s α = 0.92 overall; subscales α = 0.89–0.92), supporting strong reliability. These domains were subsequently used to generate composite indices in the current analysis, while additional RISE awareness items were added only at post-implementation time points (see Appendix A).

### 2.5. Randomization

All healthcare workers received scan tags with QR codes that linked to the ReadyWorks platform for RISE program information. A random subset of scan tags was programmed to display an invitation to complete the baseline survey, ensuring probabilistic sampling across the workforce. The anonymous survey required approximately 10 min to complete, and participants received a $10 online gift card upon completion. Two follow-up survey rounds were conducted after implementation using the same randomized scan-tag activation strategy. Because surveys were anonymous, individual responses could not be linked across waves, resulting in a repeated cross-sectional design with potential but unmeasurable participant overlap.

### 2.6. Statistical Analysis

Descriptive statistics were used to summarize demographic and work-related characteristics at baseline and follow-up time points. Mental health outcomes (GAD-7, CD-RISC-2, MBI-2) and culture of well-being subscales were scored as the sum of non-missing items. For primary analyses, continuous scale scores were calculated when at least one item was present to maximize statistical power and reduce bias related to complete-case exclusion. In sensitivity analyses, scale scores were alternatively coded as missing when any component item was missing to assess robustness. Additionally, all mental health scales were converted into binary clinical endpoints using established cut-points to enhance interpretability and permit clinically meaningful risk modeling. Missing scale items were handled differently based on analytic purpose: continuous models treated incomplete scale scores as missing to preserve psychometric validity and avoid biased score inflation, while binary endpoints used established clinical cut-points to retain partially complete cases and maximize statistical power for clinically interpretable comparisons.

Linear regression models were applied to continuous outcomes and logistic regression models to binary endpoints, comparing post-implementation time points to baseline. Robust variance estimation was selected instead of mixed-effects modeling because survey participation was strictly anonymous and respondents could not be linked across time points [30]. As a result, repeated-measures clustering by individual could not be specified, making mixed-effects models methodologically inappropriate. Robust variance estimators provided valid inference under potential within-site correlation while preserving anonymity constraints.

Models were adjusted for site, time point, age, and employment duration. Interaction terms between site and time were evaluated and retained based on Akaike Information Criterion [31]. Awareness and utilization of RISE were evaluated using chi-square testing. All analyses were conducted using Stata 18.0 [32].

## 3. Results

### 3.1. Participants Characteristics

Participant characteristics at baseline and follow up points are summarized in Table 1. Participants were broadly similar at two follow-up points, except for slightly lower proportion of respondents at the second follow-up aged 18 to 29 years of age and lower proportion of employment duration of 2 or fewer years. The overall response rate for baseline survey was 53.5% and 58.5% and 54%, respectively, for follow up 1 and 2.

### 3.2. RISE Implementation

SRH: The baseline survey was conducted between 3 August 2022 and 19 August 2022. Out of the 150 randomly selected employees, 80 (53%) completed the baseline survey. The leadership training and co-development and training included 93 leaders. The leadership team identified an educator to lead program implementation. Twenty-eight staff members completed the peer responder training. The RISE Program launched 2 months after the peer responder training. A follow up survey was sent to 300 employees, between 3 November 2023 and 18 November 2023 which was completed by 189 (63%). A second survey was sent between 15 December 2024 and 14 January 2025 to 382 employees and 219 (57%) completed the survey.

BH: A baseline survey was conducted between 13 December 2023 and 4 January 2024 completed by 160 employees out of 300 (53%). The leadership co-development and training included 69 leaders. The Chief Wellness Officer and the Wellness Committee co-led the implementation, and 213 staff members completed peer responder training. The RISE Program launched 2 months after training. This site has implemented and successfully sustained the traditional model of RISE with minimal to no adaptations. The post survey was offered between 15 December 2024 and 14 January 2025 and was completed by 220 out of 401 employees (55%).

### 3.3. Anxiety, Burnout and Resilience Outcomes

At baseline, healthcare workers reported moderate levels of anxiety. (Table 1) Bu1nout levels were elevated, with substantial proportion of respondents experiencing burnout at least weekly. Resilience scores were comparatively lower, suggesting limited capacity to adapt to stress.

Linear regression analyses indicated anxiety scores declined significantly at Time 2 (β = −1.08, 95% CI [−2.08, −0.07]), indicating reduced anxiety symptoms after two years. Employees with 3–10 years of work duration reported significantly higher anxiety scores (β ≈ 1.5–1.7, *p* < 0.05–0.01), while older staff demonstrated lower anxiety. Specifically, employees aged 45–64 and ≥65 reported approximately two-point reductions compared to those aged 18–29. Logistic regression models showed lower odds of anxiety symptoms at Time 2 (OR range 0.66–0.82), (Time 1 OR = 0.82, 95% CI 0.58–1.18; Time 2 OR = 0.66, 95% CI 0.41–1.07). Odds Ratio (OR) is a measure that shows how much the odds of an outcome increase or decrease with a given exposure. Intermediate tenure (3–5 years) was associated with increased odds of anxiety relative to <1 year (OR ≈ 1.9–2.0, *p* < 0.05). Age remained protective across models, with markedly reduced odds among employees aged 45–64 (OR ≈ 0.40, *p* < 0.001) and ≥65 (OR ≈ 0.45, *p* < 0.05).

Burnout scores declined significantly at Time 2 (β ≈ −0.70 to −0.81, *p* < 0.05). Staff at SRH reported consistently higher burnout compared to BH (β ≈ 0.57–0.66, *p* < 0.05). Intermediate work duration (3–10 years) was strongly associated with higher burnout scores (β ≈ 1.3–1.6, *p* < 0.001). Older employees showed greater resilience, and significantly lower burnout scores among those aged 45–64 (β ≈ −1.22, *p* < 0.01) and ≥65 (β ≈ −1.46, *p* < 0.01). Logistic models confirmed these patterns, with significantly increased odds of burnout for employees with 3–5 years (OR ≈ 2.3–2.6, *p* < 0.01), 6–10 years (OR ≈ 2.0–2.4, *p* < 0.05), and >10 years of tenure (OR ≈ 1.4–2.05, *p* < 0.05). In contrast, age remained protective, with reduced odds among those aged 45–64 (OR ≈ 0.42, *p* < 0.01) and ≥65 (OR ≈ 0.39, *p* < 0.05).

Resilience scores increased significantly at Time 1 (β ≈ +0.93–0.96, *p* < 0.001), though this effect was not sustained at Time 2. There was a significant site–time interaction, with SRH employees showing lower resilience at Time 1 (β ≈ −0.85, *p* < 0.001). Logistic regression confirmed reduced odds of low resilience at Time 1 (OR ≈ 0.38–0.42, *p* < 0.05), but this was not maintained at Time 2. The site–time interaction persisted, with SRH employees exhibiting higher odds of low resilience (interaction OR ≈ 3.0, *p* < 0.05). Employment duration and age were not consistently associated with resilience.

This interaction indicates that the impact of the RISE program on resilience differed by site and must be interpreted within each organizational context rather than as a uniform overall effect. Specifically, while both hospitals demonstrated initial gains in resilience, the direction and magnitude of change varied: Bayhealth employees showed significant short-term improvement in resilience following implementation, whereas Shore Regional Health (SRH) staff exhibited comparatively smaller gains at the same time point. This divergence suggests that site-specific contextual factors—such as staffing resources, leadership engagement, and degree of program integration—moderated the program’s impact. Accordingly, interpretation of resilience trends should focus on within-site changes over time rather than pooled estimates across sites.

Overall, findings suggest modest reductions in anxiety and burnout, with an initial increase in resilience that was not maintained at follow-up. Resilience improved shortly after implementation (at ~1 year), but the gains were not maintained by the second follow-up (~2 years). Intermediate work duration (3–10 years) conferred higher risk for anxiety and burnout, while older age was a protective factor across models. Site-level differences were notable, with SRH employees experiencing higher burnout and lower resilience than their Bayhealth counterparts. These findings are summarized in Table 2, Table 3 and Table 4.

### 3.4. Culture of Wellbeing

The results were generally positive, with some variation across follow-up periods. Organizational support scores remained stable from baseline through both follow-ups, with no significant change (*p* = 0.79). Access to peer support improved significantly over time, from a mean of 9.8 at baseline to 10.9 at follow-up 2 (*p* = 0.007). Similarly, the availability of support subscale improved significantly, with significant differences across time points (*p* = 0.002). Taken together, while perceived organizational support did not change, perceived opportunities for peer support and perceptions of support availability increased over the follow-up periods. These findings are summarized in Figure 1.

### 3.5. Awareness of RISE

The post implementation RISE evaluation indicated increasing awareness and utilization over time, with notable differences by site. At the first follow-up, 42.7% of Bayhealth staff and 36.5% of SRH staff reported having heard of RISE, but by the second follow-up at SRH, awareness rose substantially to 66.7% (*p* < 0.001). Awareness was gained through multiple channels, including staff meetings, word of mouth, brochures, email, and posters. RISE utilization was greater at SRH between follow-ups among respondents who reported having heard of RISE, from 7.7 to 38.5%, suggesting not only greater recognition of RISE but also a growing integration into staff support practices. This change did not reach statistical significance (*p* = 0.203). Overall, these findings highlight the feasibility of the RISE peer support program in rural hospital sites.

The increase in RISE awareness and utilization over time likely reflects the cumulative impact of multi-channel dissemination strategies, including leadership endorsement during staff meetings, targeted email messaging, posters in clinical areas, peer champion advocacy, and informal word-of-mouth diffusion.

### 3.6. Contextual Differences

Between Sites: SRH and BH differed substantially in implementation scale and institutional capacity. SRH required compressed peer responder training and tailored leadership co-development due to staffing constraints and resource limitations, whereas BH implemented the traditional RISE model with minimal adaptation and a larger training cohort. These contextual differences likely contributed to the observed site-level variation in burnout and resilience outcomes and underscore that implementation conditions significantly shape program impact in rural healthcare systems.

## 4. Discussion

This study is the first to evaluate the implementation of the Johns Hopkins RISE peer support model in rural hospital settings, addressing a critical gap in the literature on burnout and resilience interventions outside of academic medical centers and large health systems. While peer support programs have demonstrated effectiveness in tertiary care institutions [33,34,35,36,37,38,39], their feasibility, effectiveness, and acceptability in resource-limited rural environments had not been evaluated. By adapting and implementing the RISE program in two rural hospitals, this study provides evidence that such interventions are not only feasible but also associated with modest improvements in anxiety, burnout, and resilience outcomes, after a short period of implementation, along with meaningful gains in perceived peer support and availability of supportive resources. This is important because health worker perceptions of support have been shown to be related to institutional safety culture and workforce wellbeing [40].

The strength of this work lies in the comparative insights generated across two rural hospital systems with distinct implementation approaches. At SRH, the program required substantial modification, including compressed responder training and leadership co-development tailored to organizational capacity. In contrast, BH implemented the traditional model with minimal adaptation. These differences yielded interesting observations: SRH employees experienced higher burnout and lower resilience relative to their BH counterparts, underscoring how contextual and organizational factors influence program effectiveness. This site-level variation provides empirical evidence that implementation context may be important in determining outcomes in rural healthcare systems.

The study also highlights the role of workforce tenure and age in shaping intervention impact. Staff with 3–10 years of service consistently reported greater anxiety and burnout, suggesting that mid-career employees may represent a particularly vulnerable subgroup in rural hospitals. Conversely, older age appeared to convey protective effects against both anxiety and burnout, a finding that underscores the value of examining developmental and career-stage differences when designing interventions to support resilience. Reasons for this might include demand/resource imbalance [41] or age-related emotion-regulation advantages [42]. The elevated anxiety and burnout observed among staff with intermediate tenure (3–10 years) may reflect a period of heightened vulnerability marked by growing clinical responsibility, leadership expectations, and cumulative exposure to moral distress without the job control, career security, or institutional influence that often accompany senior roles. This misalignment between workload demands and available support or autonomy may intensify distress during this career stage. Future interventions could specifically target this group through structured mentorship, leadership development pathways, protected time for peer support participation, and role-specific resilience training to mitigate burnout and promote retention during this high-risk transition period. To our knowledge, few studies have linked these workforce factors with peer support program outcomes, making this an important contribution.

Beyond individual outcomes, this study contributes insights into organizational culture change in rural hospitals. Although perceptions of organizational support perceptions were stable, access to peer support and availability of supportive resources improved significantly over time. These findings suggest the potential for peer support interventions to enhance the structural capacity for resilience in rural hospitals even without broader organizational culture shifts. While resilience gains were not sustained at the second follow-up, it may be more likely that instituting organizational structures for peer support helps increase individual resilience among healthcare workers. Our findings suggest that resilience is dynamic and may require continued reinforcement and institutional support to be maintained over time

These results are consistent with prior research highlighting the effectiveness of peer support interventions in mitigating burnout and enhancing psychological well-being in high-stress healthcare environments [42]. The tailored adaptation of the RISE model to a rural context underscores the importance of systemic, multi-level interventions that combine individual resilience-building with organizational culture to address burnout.

### Limitations

This study was part of a larger protocol and has several limitations. The study was limited to two sites and was not randomized. This may diminish the power to detect significant differences. The pre-post design without a concurrent control group limits causal inference. Survey participation, while moderate, varied across sites and time points, raising the possibility of selection bias. Reliance on self-reported measures may have introduced reporting bias, particularly given the sensitive nature of burnout and mental health outcomes. Although validated scales were used for key domains, some new items developed to assess perceptions of resilience and support had not been previously validated, which may affect generalizability and reproducibility. Although the two-item versions of the CD-RISC and MBI demonstrate strong validity for screening [27], their sensitivity to detect subtle or sustained changes over time may be lower than that of the full-length instruments, potentially attenuating observed longitudinal effects. Because survey responses were anonymous and could not be linked across time points, we were unable to account for repeated measures at the individual level, which limits causal inference and the precision of longitudinal effect estimates. Finally, findings reflect the experiences of two rural hospital systems in the Mid-Atlantic region, which may not capture the diversity of rural healthcare settings across the U.S. or globally.

## 5. Conclusions

In summary, these results highlight that a peer support program such as RISE can be successfully adapted to rural contexts, fostering resilience and reducing burnout among healthcare workers while strengthening institutional stockpile of workforce well-being. By demonstrating feasibility, acceptability, and measurable impact in rural hospitals, this study expands the evidence base for resilience interventions into underserved settings where burnout threatens both workforce retention and patient care access. Future research should expand to diverse rural settings with larger, randomized samples to validate these outcomes. Longitudinal studies could assess the long-term effects of peer support programs on turnover rates and patient care quality. By addressing these gaps, healthcare systems can better implement evidence-based strategies to support rural healthcare workers, ultimately strengthening workforce stability and improving care delivery in underserved communities.

## Figures and Tables

**Figure 1 healthcare-14-00091-f001:**
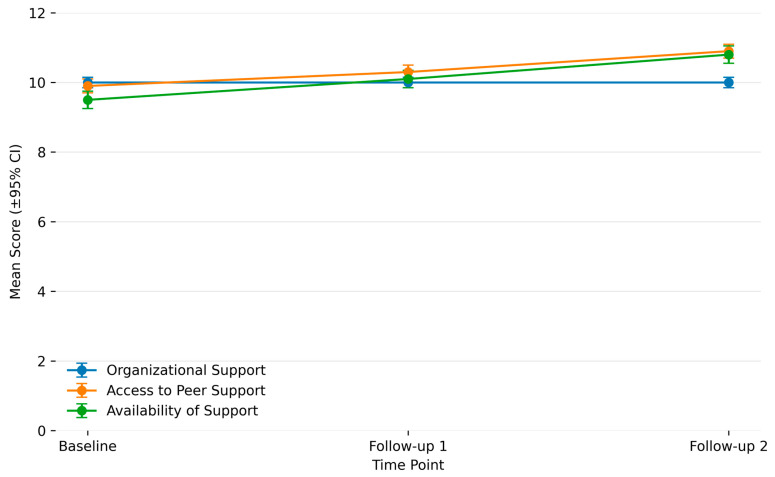
Change In Culture of Wellbeing Scores Over Time With 95% CI. Note: While Access to Peer Support and Availability of Support increased significantly over time, Organizational Support scores remained stable, indicating that gains reflected improved peer-level support rather than broader changes in institutional culture during the study period.

**Table 1 healthcare-14-00091-t001:** Rural Hospital Participant Characteristics by Time Point.

	Baseline	Follow-Up 1	Follow-Up 2
	*n* = 240	*n* = 409	*n* = 219
Age category (years)			
18–29	31 (12.9%)	47 (11.5%)	15 (6.8%)
30–44	82 (34.2%)	114 (27.9%)	74 (33.8%)
45–64	104 (43.3%)	161 (39.4%)	86 (39.3%)
65 and older	14 (5.8%)	28 (6.8%)	17 (7.8%)
Missing	0 (0.0%)	38 (9.3%)	14 (6.4%)
Prefer not to answer	9 (3.8%)	21 (5.1%)	13 (5.9%)
Race category			
White	158 (65.8%)	244 (59.7%)	144 (65.8%)
African American	35 (14.6%)	61 (14.9%)	28 (12.8%)
Asian	9 (3.8%)	7 (1.7%)	1 (0.5%)
Native Hawaiian	1 (0.4%)	0 (0.0%)	0 (0.0%)
American Indian or Alaska Native	1 (0.4%)	0 (0.0%)	1 (0.5%)
Multiple Races	7 (2.9%)	3 (0.7%)	1 (0.5%)
Prefer not to answer	29 (12.1%)	94 (23.0%)	44 (20.1%)
Latino Ethnicity			
No	215 (89.6%)	311 (76.0%)	178 (81.3%)
Yes	11 (4.6%)	15 (3.7%)	5 (2.3%)
Prefer not to answer	14 (5.8%)	45 (11.0%)	21 (9.6%)
Missing	0 (0.0%)	38 (9.3%)	15 (6.8%)
Gender			
Female	188 (78.3%)	288 (70.4%)	170 (77.6%)
Male	35 (14.6%)	49 (12.0%)	18 (8.2%)
Trans	1 (0.4%)	0 (0.0%)	0 (0.0%)
Non-Binary	0 (0.0%)	2 (0.5%)	0 (0.0%)
Else	1 (0.4%)	0 (0.0%)	0 (0.0%)
Prefer to not Answer	15 (6.2%)	32 (7.8%)	17 (7.8%)
Missing	0 (0.0%)	38 (9.3%)	14 (6.4%)
Sexual Orientation			
Gay/Lesbian	9 (3.8%)	5 (1.2%)	0 (0.0%)
Straight	211 (87.9%)	318 (77.8%)	177 (80.8%)
Bisexual	4 (1.7%)	4 (1.0%)	0 (0.0%)
Other	1 (0.4%)	1 (0.2%)	1 (0.5%)
Prefer not to answer	15 (6.2%)	42 (10.3%)	25 (11.4%)
Missing	0 (0.0%)	39 (9.5%)	16 (7.3%)
Employment Duration			
<1 year	33 (13.8%)	41 (10.0%)	13 (5.9%)
1–2 years	35 (14.6%)	48 (11.7%)	26 (11.9%)
3–5 years	43 (17.9%)	65 (15.9%)	37 (16.9%)
6–10 years	42 (17.5%)	62 (15.2%)	35 (16.0%)
>10 years	80 (33.3%)	133 (32.5%)	83 (37.9%)
Prefer not to answer	7 (2.9%)	22 (5.4%)	11 (5.0%)
Missing	0 (0.0%)	38 (9.3%)	14 (6.4%)
Position			
Clinician/Provider	89 (37.1%)	125 (30.6%)	71 (32.4%)
Technician/Therapist	63 (26.2%)	94 (23.0%)	63 (28.8%)
Leadership/Administration	19 (7.9%)	27 (6.6%)	14 (6.4%)
Receptionist/Office Assistant	22 (9.2%)	31 (7.6%)	20 (9.1%)
Other/Non-patient Contact	28 (11.7%)	42 (10.3%)	12 (5.5%)
Missing	19 (7.9%)	90 (22.0%)	39 (17.8%)

**Table 2 healthcare-14-00091-t002:** Results of Linear regression models for Mental Health and Culture of Well-being Outcomes.

	Anxiety Score		Burnout Score		Resilience Score		Culture of Well-Being Total Score	
	Estimated Slope	95% Confidence Interval	Estimated Slope	95% Confidence Interval	Estimated Slope	95% Confidence Interval	Estimated Slope	95% Confidence Interval
Follow-up Time								
Time 1 vs. Baseline	−0.479	[−1.277,0.319]	−0.270	[−0.762,0.223]	0.958 ***	[0.710,1.206]	1.675 *	[0.0237,3.326]
Time 2 vs. Baseline	−1.077 *	[−2.084,−0.0705]	−0.807 *	[−1.469,−0.144]	0.0415	[−0.301,0.384]	4.585 ***	[2.385,6.785]
Site								
SRH vs. Bayhealth	0.306	[−0.479,1.091]	0.663 *	[0.155,1.170]	−0.0640	[−0.421,0.293]	−3.934 ***	[−5.648,−2.219]
Interaction Term								
Time 1 × SRH					−0.859 ***	[−1.285,−0.433]		
Employment Duration ^@^								
1–2 years vs. <1 year	1.068	[−0.292,2.428]	1.139 **	[0.305,1.974]	−0.0701	[−0.455,0.315]	−3.052 *	[−6.092,−0.0118]
3–5 years vs. <1 year	1.705 **	[0.410,2.999]	1.550 ***	[0.773,2.326]	−0.254	[−0.614,0.106]	−4.461 **	[−7.334,−1.589]
6–10 years vs. <1 year	1.424 *	[0.0602,2.789]	1.621 ***	[0.857,2.386]	−0.171	[−0.538,0.196]	−3.750 ^*^	[−6.636,−0.864]
>10 years vs. <1 year	0.909	[−0.277,2.095]	1.339 ***	[0.613,2.065]	0.0944	[−0.251,0.440]	−2.613	[−5.355,0.129]
Age ^#^								
30–44 vs. 18–29	−0.807	[−2.050,0.436]	−0.539	[−1.317,0.239]	−0.0713	[−0.379,0.237]	−1.360	[−3.753,1.033]
45–64 vs. 18–29	−2.092 **	[−3.345,−0.839]	−1.222 **	[−2.005,−0.439]	0.122	[−0.190,0.434]	−0.0776	[−2.494,2.339]
65 and older vs. 18–29	−2.032 *	[−3.735,−0.330]	−1.456 **	[−2.462,−0.451]	0.104	[−0.310,0.519]	1.486	[−1.633,4.605]
Intercept	4.933 ***	[3.552,6.313]	5.450 ***	[4.650,6.250]	6.745 ***	[6.362,7.127]	42.00 ***	[39.51,44.48]
F statistic (*p*-value)	F(10, 742) = 3.64 (<0.001)		F(10, 742) = 4.33 (<0.001)		F(11, 741) = 12.94 (<0.001)		F(10, 742) = 4.44 (<0.001)	
R-squared	0.043		0.049		0.121		0.057	
Observations	753		753		753		753	

**^@^** 92 participants have missing data on age, including 40 who preferred not to answer; **^#^** 95 participants have missing data on age, including 43 preferred not to answer; * *p* < 0.05, ** *p* < 0.01, *** *p* < 0.001.

**Table 3 healthcare-14-00091-t003:** Results of Multivariable Logistic regression models for Mental Health Outcomes.

	Anxiety ^a^		Burnout ^b^		Low Resilience ^c^	
	Estimated Odds Ratio	95% Confidence Interval	Estimated Odds Ratio	95% Confidence Interval	Estimated Odds Ratio	95% Confidence Interval
Follow-up Time						
Time 1 vs. Baseline	0.831	[0.583,1.184]	0.850	[0.586,1.233]	0.376 *	[0.176,0.804]
Time 2 vs. Baseline	0.664	[0.412,1.069]	0.630	[0.383,1.036]	1.338	[0.589,3.041]
Site						
SRH vs. Bayhealth	1.333	[0.929,1.912]	1.350	[0.931,1.957]	0.877	[0.375,2.046]
Interaction Term						
Time 1 × SRH					3.183 *	[1.004,10.09]
Employment Duration						
1–2 years vs. <1 year	1.468	[0.800,2.695]	1.199	[0.595,2.414]	0.931	[0.393,2.204]
3–5 years vs. <1 year	2.046 *	[1.137,3.683]	2.563 **	[1.348,4.875]	1.112	[0.504,2.454]
6–10 years vs. <1 year	1.529	[0.834,2.801]	2.386 *	[1.222,4.656]	0.985	[0.432,2.247]
>10 years vs. <1 year	1.656	[0.934,2.939]	2.053 *	[1.076,3.916]	0.567	[0.255,1.263]
Age						
30–44 vs. 18–29	0.807	[0.481,1.354]	0.732	[0.426,1.256]	1.245	[0.598,2.591]
45–64 vs. 18–29	0.397 ***	[0.232,0.676]	0.423 **	[0.239,0.748]	1.001	[0.472,2.124]
65 and older vs. 18–29	0.446 *	[0.213,0.933]	0.385 *	[0.174,0.855]	1.223	[0.414,3.614]
Intercept	753		753		753	
AIC	998.9		912.1		568.9	

^a^ Anxiety is defined as mild, moderate or severe or anxiety score 5 and above on GAD-7; ^b^ Burnout is defined as “once a week”, “few times a week” or “every day” response to one of two questions on the Maslach Burnout Inventory (MBI). ^c^ Low resilience was defined as a score lower than 6 on the sum of the two items on the CD-RISC scale. Exponentiated coefficients; 95% confidence intervals in brackets; * *p* < 0.05, ** *p* < 0.01, *** *p* < 0.001.

**Table 4 healthcare-14-00091-t004:** Mean (standard deviation) scores for mental health and culture of well-being at baseline and follow-up time points, with corresponding effect sizes (Cohen’s d).

Outcome	Baseline (*n* = 240) Mean (SD)	Follow-Up 1 (*n* = 409) Mean (SD)	Follow-Up 2 (*n* = 219) Mean (SD)	Cohen’s d Baseline vs. F/U 1	Cohen’s d Baseline vs. F/U 2
GAD-7 Anxiety Score	4.65 (4.93)	4.28 (4.61)	4.11 (4.49)	0.08	0.12
MBI-2 Burnout Score	5.98 (2.87)	5.94 (3.01)	5.88 (2.92)	0.01	0.04
CD-RISC-2 Resilience Score	6.70 (1.30)	7.23 (1.21)	6.71 (1.24)	−0.42	−0.01
Organizational Support	9.03 (2.04)	9.15 (2.33)	8.97 (2.05)	−0.05	0.03
Access to Peer Support	9.80 (3.60)	10.40 (3.67)	10.86 (3.36)	−0.16	−0.30
Availability of Support	18.48 (5.68)	18.57 (5.92)	18.89 (5.64)	−0.02	−0.07
Culture of Well-Being (Total)	37.31 (9.60)	37.67 (10.95)	38.40 (9.86)	−0.03	−0.11

Note: Values are presented as mean (standard deviation). Cohen’s d values represent standardized mean differences comparing baseline with each follow-up time point. Positive values indicate improvement for anxiety and burnout (lower scores), whereas negative values indicate improvement for resilience and culture-of-well-being measures (higher scores). Effect sizes are interpreted as small (0.2), moderate (0.5), and large (0.8).

## Data Availability

The raw data supporting the conclusions of this article will be made available by the authors on request. The data are not publicly available due to privacy and ethical restrictions.

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
