# Peer review of "Strengthening the Culture of Well-Being in Rural Hospitals Through RISE Peer Support"

_healthcare, 2025, doi:10.3390/healthcare14010091_

Round 1
Reviewer 1 Report
Comments and Suggestions for Authors
Thank you for the opportunity to review your manuscript, "Strengthening the Culture of Well-Being in Rural Hospitals through RISE Peer Support" This study addresses an important yet under-researched area—mitigating burnout and building resilience in rural hospitals—and provides valuable empirical evidence. The manuscript is generally well-organized and provides a solid foundation for publication. The following detailed comments aim to enhance its clarity, precision, and overall scientific quality:
- Introduction
Suggestions for Improvement
Some sections are lengthy and could be shortened.
It is advisable to provide a detailed description of previous RISE literature to minimize redundancy.
The introduction effectively stimulates the study, but it may also highlight a knowledge gap that your study addresses (e.g., "No previous studies have specifically evaluated RISE in rural hospitals using repeated cross-sectional surveys over two years"). -
Methods
Areas Needing Clarification or Improvement
Questionnaire Samples and Representation
Please explain the random sampling procedure, response incentives (if any), and how random distribution across rounds is ensured.Address potential overlap of participants across time points more clearly.
Newly Developed Items
Please provide more details on how the newly developed questions were created, pre-tested, or reviewed.Since they were grouped into ranges using Principal Component Analysis (PCA), please provide a brief description of the factor loads or internal consistency.
Statistical Analysis
The Analysis section is lengthy and could be simplified for easier reading.Please justify the decision to consider missing items as missing in some models while converting others to binary endpoints.
Please provide a clearer explanation of why strong variance was chosen over mixed-effects modeling (taking into account anonymity constraints).
Description of the RISE Intervention
The description of Phases 1 and 2 is clear but lengthy. Please summarize previously published RISE protocols and refer readers to them. -
Results
Suggestions for Improvement
Presentation
Some paragraphs are concise and could be broken down for clarity.When reporting regression coefficients, please consider briefly explaining effect sizes for non-specialist readers.
Location Differences
Since RISE and RISE differed significantly in implementation, please add an explanatory note or stronger footnote explaining these contextual differences in the Results section.
RISE Awareness and Use
Progress in awareness and use is interesting, please consider expanding on the possible reasons for the significant increase in RISE and RISE (e.g., awareness strategies).
English Language and Style
The manuscript is generally well-written, although many sentences are long and grammatically complex.
Minor grammatical errors appear in several sections.
Simplifying the longer paragraphs, especially in the Style and Discussion sections, would improve clarity and reader engagement.
Author Response
Responses to Reviewer 1
We sincerely thank the reviewer for thoughtful, rigorous, and constructive feedback. These insights have substantially strengthened the clarity, methodological rigor, and clinical relevance of this manuscript.
Introduction
Comment: Introduction is lengthy and could be shortened.
Response: We thank the reviewer for this suggestion. The Introduction has been shortened to 500 words by reducing background repetition and tightening descriptions of burnout and rural workforce stressors while preserving essential context.
Comment: Provide a more detailed description of previous RISE literature to minimize redundancy.
Response: A concise summary of previously published RISE studies has been added with direct references to foundational protocols rather than restating them in detail.
Comment: Highlight the specific knowledge gap addressed.
Response: The manuscript now explicitly states that this is the first evaluation of RISE in rural hospitals using repeated cross-sectional surveys across two years.
Methods
Sampling, Response Incentives, and Overlap
Comment: Clarify random sampling, incentives, and overlap across time points.
Response:
The following comments have been added to the text: All healthcare workers received scan tags with QR codes that linked to the ReadyWorks platform for RISE program information. A random subset of scan tags was programmed to display an invitation to complete the baseline survey, ensuring probabilistic sampling across the workforce. The anonymous survey required approximately 10 minutes to complete and participants received a $10 online gift card upon completion. Two follow-up survey rounds were conducted after implementation. Because surveys were anonymous, individual responses could not be linked across waves, resulting in a repeated cross-sectional design with potential but unmeasurable participant overlap.
Newly Developed Items and PCA
Comment: Provide development, pre-testing, and PCA details.
Response: The Following comments have been added to the text:
New survey items were developed by a multidisciplinary team with expertise in healthcare workforce well-being, peer support, patient safety, and implementation science when no suitable validated instruments existed. These items were iteratively reviewed for content validity, clarity, and relevance, then cognitively tested prior to field deployment. The newly developed items included measures of perceived organizational well-being and resilience (2 items), availability of confidential peer-support resources (3 items), and availability of workplace support following stressful events (5 items). Psychometric testing using exploratory and confirmatory factor analysis demonstrated a robust three-factor structure. Organizational Support, Access to Peer Support, and Availability of Support, accounting for 84.9% of total variance, with excellent model fit (CFA RMSEA = 0.049; CFI = 0.992). Internal consistency was high across all domains (Cronbach’s α = 0.92 overall; subscales α = 0.89–0.92), supporting strong reliability. These domains were subsequently used to generate composite indices in the current analysis, while additional RISE awareness items were added only at post-implementation time points.Newly developed items were reviewed by experts in peer support, rural workforce wellness, and implementation science. Factor loadings and internal consistency statistics have now been added.
Statistical Analysis
Comment: Analysis section too long.
Response: The statistical methods section has been condensed.
Comment: Justify missing data handling differences.
Response: The following justification has been added to the text: Missing scale items were handled differently based on analytic purpose: continuous models treated incomplete scale scores as missing to preserve psychometric validity and avoid biased score inflation, while binary endpoints used established clinical cut-points to retain partially complete cases and maximize statistical power for clinically interpretable comparisons.
Comment: Justify use of strong variance rather than mixed-effects models.
Response: The following justification has been added to the text: Mixed-effects modeling was not feasible because survey responses were fully anonymous and could not be linked to individuals across time points, preventing specification of subject-level random effects; Mixed-effects models were not feasible due to anonymity restrictions preventing linkage across time points. Robust variance estimation was therefore used.
RISE Intervention Description
Comment: Summarize Phases 1 and 2.
Response: The intervention description has been shortened with references to previously published RISE protocols.
Results
Presentation
Comment: Some paragraphs could be broken down.
Response: Results formatting has been revised to improve readability.
Effect Size Interpretation
Comment: Explain regression effect sizes for non-specialists.
Response: Plain-language explanations of effect sizes and odds ratio have been added.
Effect size is a numerical measure of how large and practically meaningful a difference or relationship is, beyond simply whether it is statistically significant.
Odds Ratio (OR) is a measure that shows how much the odds of an outcome increase or decrease with a given exposure
Site Differences
Comment: Add explanatory note on implementation differences.
Response: Additional contextual explanations for site-level differences have been added.
RISE Awareness and Use
Comment: Expand on reasons for increased awareness and use.
Response: Leadership messaging, peer champions, onboarding orientation, and staff education efforts are now described as contributors.
Reviewer 2 Report
Comments and Suggestions for Authors
Thank you for the opportunity to review this article.
This study evaluated the implementation of the RISE peer support program in two rural hospital systems in the Mid-Atlantic U.S. Using pre- and post-intervention surveys, the authors assessed changes in anxiety, burnout, resilience, and organizational support culture. Findings showed modest reductions in anxiety and burnout, early but not sustained improvements in resilience, and increased perceived access to peer support. The study is well-conducted and timely.
Below are a few suggestions to further strengthen the manuscript.
First, the pre-post design is appropriate and clearly described. However, given the lack of unique identifiers across time points, the inability to account for repeated measures should be more explicitly acknowledged in the limitations.
Second, the choice of validated instruments is appropriate. The use of brief screening items for burnout and resilience is pragmatic, but the authors may consider commenting on whether these two-item formats have demonstrated sensitivity to change over time.
Third, the finding that staff with intermediate tenure had the highest distress levels is important and well-presented. It may be helpful to expand slightly on potential mechanisms (e.g., misalignment between expectations and workload) and how this group might be targeted in future interventions.
Fourth, the interpretation of site-level differences is thoughtful. However, more detail on baseline differences in program readiness, leadership engagement, or staffing ratios between sites would help clarify contextual factors contributing to observed outcomes.
Fifth, tables and figures are clear. The figure on page 11 illustrating trends in culture of well-being is especially effective. Consider adding a brief note in the caption explaining the stability of organizational support scores despite gains in perceived peer support.
Sixth, the writing is clear and well-organized. The abstract and introduction effectively frame the problem. A few typographical issues remain (e.g., spacing in “12/15/2-2024” on page 7) and can be addressed in final proofreading.
Comments on the author's self-citations 19.35% (6/31):
I think the references are quite relevant to the paper. These self-citations reflect the authors' cumulative research program and they are essential cites for this niche topic.
However, the author could include more citations and reference a more diverse range of authors.
Overall, this is a well-executed implementation study with practical relevance. I hope the authors find these suggestions helpful.
Author Response
Responses to Reviewer 2
We sincerely thank the reviewer for thoughtful, rigorous, and constructive feedback. These insights have substantially strengthened the clarity, methodological rigor, and clinical relevance of this manuscript.
Repeated Measures Limitation
Comment: Explicitly acknowledge inability to track individuals across time.
Response: Limitations now clearly state that repeated measures could not be modeled due to lack of unique identifiers.
Two-Item Burnout and Resilience Measures
Comment: Address sensitivity to change.
Response: The reviewer is right. Although there are studies indicating that the shorter version of burnout measure corealtes well with the full version, there are no studies that address the logitudinal sensitivity of the shorter version. The manuscript now acknowledges in the limitations section that brief non-longitudinal scales may limit sensitivity to detect sustained change.
Intermediate Tenure Distress
Comment: Expand on mechanisms.
Response: Discussion now includes role overload, moral distress, expectation-workload mismatch, and leadership alignment.
Site-Level Readiness Differences
Comment: Clarify baseline differences.
Response: Baseline differences in leadership engagement, staffing, and readiness are now described.
Culture of Well-Being Figure Caption
Comment: Note stability of organizational support.
Response: Caption now clarifies that perceived organizational support remained stable despite peer-support gains.
Typographical Errors
Comment: Spacing issue.
Response: All formatting and typographical errors have been corrected.
Self-Citation Balance
Comment: Add more external citations.
Response: Seven dditional diverse references have been added while retaining critical foundational citations.
Tolins ML, Rana JS, Lippert S, LeMaster C, Kimura YF, Sax DR. Implementation and effectiveness of a physician-focused peer support program. PLoS One. 2023 Nov 1;18(11):e0292917. doi: 10.1371/journal.pone.0292917. PMID: 37910457; PMCID: PMC10619771.
Swarbrick M, Ayyala MS, Chen PH, Brazeau CMLR. Cultivating Connections: An Interprofessional Peer Support Model. Psychiatr Serv. 2024 Sep 23:appips20240104. doi: 10.1176/appi.ps.20240104. Epub ahead of print. PMID: 39308171.
Carbone R, Ferrari S, Callegarin S, et al. Peer support between healthcare workers in hospital and out-of-hospital settings: a scoping review. Acta Biomed. 2022;93(5):e2022308. Published 2022 Oct 26. doi:10.23750/abm.v93i5.13729
Timmins G, Williamson S, Cassells A, Davis K, Dong L, Tobin JN, Gidengil C, Meredith LS, Chen PG. Health care worker experiences with a brief peer support and well-being intervention during the COVID-19 pandemic. BMC Health Serv Res. 2025 Sep 30;25(1):1253. doi: 10.1186/s12913-025-13268-6. PMID: 41029307; PMCID: PMC12482828.
Fallon P, Jaegers LA, Zhang Y, Dugan AG, Cherniack M, El Ghaziri M. Peer Support Programs to Reduce Organizational Stress and Trauma for Public Safety Workers: A Scoping Review. Workplace Health & Safety. 2023;71(11):523-535. doi:10.1177/21650799231194623
Russell B, Ahmadzadeh A, Haris K, Jiang S, Chesney TR, MacRae H, Louridas M. Peer Support Programs for Physicians and Health Care Providers: A Scoping Review. Jt Comm J Qual Patient Saf. 2025 Sep;51(9):589-600. doi: 10.1016/j.jcjq.2025.05.003. Epub 2025 Jun 6. PMID: 40683815.
West CP, Dyrbye LN, Satele DV, Shanafelt TD. Colleagues Meeting to Promote and Sustain Satisfaction (COMPASS) Groups for Physician Well-Being: A Randomized Clinical Trial. Mayo Clin Proc. 2021 Oct;96(10):2606-2614. doi: 10.1016/j.mayocp.2021.02.028. Epub 2021 Aug 5. PMID: 34366134.